# Pan-Genotypic Direct-Acting Antiviral Agents for Undetermined or Mixed-Genotype Hepatitis C Infection: A Real-World Multi-Center Effectiveness Analysis

**DOI:** 10.3390/jcm11071853

**Published:** 2022-03-27

**Authors:** Hsu-Heng Yen, Yang-Yuan Chen, Jun-Hung Lai, Hung-Ming Chen, Chih-Ta Yao, Siou-Ping Huang, I-Ling Liu, Ya-Huei Zeng, Fang-Chi Yang, Fu-Yuan Siao, Mei-Wen Chen, Pei-Yuan Su

**Affiliations:** 1Division of Gastroenterology, Department of Internal Medicine, Changhua Christian Hospital, Changhua 500, Taiwan; 91646@cch.org.tw (H.-H.Y.); 27716@cch.org.tw (Y.-Y.C.); 182972@cch.org.tw (S.-P.H.); 125267@cch.org.tw (I.-L.L.); 120693@cch.org.tw (Y.-H.Z.); 84156@cch.org.tw (F.-C.Y.); 2Artificial Intelligence Development Center, Changhua Christian Hospital, Changhua 500, Taiwan; 3General Education Center, Chienkuo Technology University, Changhua 500, Taiwan; 4Department of Electrical Engineering, Chung Yuan University, Taoyuan 320, Taiwan; 5College of Medicine, National Chung Hsing University, Taichung 400, Taiwan; 6Division of Gastroenterology, Department of Internal Medicine, Yuanlin Christian Hospital, Changhua 500, Taiwan; 7Department of Hospitality, MingDao University, Changhua 500, Taiwan; 8Division of Gastroenterology, Department of Internal Medicine, Erhlin Christian Hospital, Changhua 500, Taiwan; 61004@cch.org.tw; 9Division of Gastroenterology, Department of Internal Medicine, Yunlin Christian Hospital, Yunlin 648, Taiwan; 820199@cch.org.tw; 10Division of Gastroenterology, Department of Internal Medicine, Lukang Christian Hospital, Changhua 500, Taiwan; 704150@cch.org.tw; 11Department of Emergency Medicine, Changhua Christian Hospital, Changhua 500, Taiwan; 57385@cch.org.tw; 12Department of Kinesiology, Health and Leisure, Chienkuo Technology University, Changhua 500, Taiwan; 13Department of Mechanical Engineering, Chung Yuan Christian University, Taoyuan 320, Taiwan; 14Department of Information Management, Chien-Kuo Technology University, Chunghua 500, Taiwan; 135442@cch.org.tw

**Keywords:** hepatitis C, direct antiviral therapy, elimination

## Abstract

Although the pan-genotypic direct-acting antiviral regimen was approved for treating chronic hepatitis C infection regardless of the hepatitis C virus (HCV) genotype, real-world data on its effectiveness against mixed-genotype or genotype-undetermined HCV infection are scarce. We evaluated the real-world safety and efficacy of two pan-genotypic regimens (Glecaprevir/Pibrentasvir and Sofosbuvir/Velpatasvir) for HCV-infected patients with mixed or undetermined HCV genotypes from the five hospitals in the Changhua Christian Care System that commenced treatment between August 2018 and December 2020. This retrospective study evaluated the efficacy and safety of pan-genotypic direct-acting antiviral (DAA) treatment in adults with HCV infection. The primary endpoint was the sustained virological response (SVR) observed 12 weeks after completing the treatment. Altogether, 2446 HCV-infected patients received the pan-genotypic DAA regimen, 37 (1.5%) patients had mixed-genotype HCV infections and 110 (4.5%) patients had undetermined HCV genotypes. The mean age was 63 years and 55.8% of our participants were males. Nine (6.1%) patients had end-stage renal disease and three (2%) had co-existing hepatomas. We lost one patient to follow-up during treatment and one more patient after treatment. A total of four patients died. However, none of these losses were due to treatment-related side effects. The rates of SVR12 for mixed-genotype and genotype-undetermined infections were 97.1% and 96.2%, respectively, by per-protocol analyses, and 91.9% and 92.7% respectively, by intention-to-treat population analyses. Laboratory adverse events with grades ≥3 included anemia (2.5%), thrombocytopenia (2.5%), and jaundice (0.7%). Pan-genotypic DAAs are effective and well-tolerated for mixed-genotype or genotype-undetermined HCV infection real-world settings.

## 1. Introduction

Chronic hepatitis C virus (HCV) infection is one of the common causes of liver cirrhosis and liver cancer that can now be prevented by effective antiviral therapy [1,2,3,4,5,6]. Over the past two decades, interferon-based therapy was the standard of care; however, its use is limited to numerous treatment-related side effects, the risk of liver decompensation among cirrhotic patients, and the need to understand the genotype of HCV to determine the treatment duration. The treatment of HCV was revolutionized with the introduction of direct-acting antiviral (DAA) therapy in recent years. This therapy was well tolerated in different treatment populations with high successful treatment rates and impressive safety profiles. These agents have been reimbursed in Taiwan since 2017, and the government of Taiwan has set the goal of obtaining an 80% treatment coverage rate with DAAs by 2025 [1,7,8,9]. There are still several barriers to HCV elimination, such as the limitation of patient access to therapy, insufficient knowledge of HCV therapy, lack of awareness of the disease, and the complexity of treatment [4,10,11,12].

During the late 2010s, several different competing DAA regimens were approved. One of the limitations of such earlier DAAs was the need to know the specific HCV genotype before initiating the therapy, i.e., elbasvir/grazoprevir did not work for patients with HCV of genotype 3 or genotype 4. Erroneous genotyping may lead to potentially suboptimal treatments with a remarkable increase in treatment costs [13]. The 2020 EASL [14] and 2019 AASLD [15,16] treatment guidelines now suggest two main regimens for treatment-naïve patients, i.e., glecaprevir (300 mg)/pibrentasvir (120 mg) (GLE/PIB), and sofosbuvir (400 mg)/velpatasvir (100 mg) (SOF/VEL) with pan-genotypic antiviral activity to simplify the treatment algorithm. A minority of the patients may get infected with a mixed HCV genotype [17,18,19] or the HCV genotype that was undetermined by conventional polymerase chain reaction (PCR) techniques [20,21]. To date, there is a paucity of real-world data on the effectiveness and safety of pan-genotypic DAAs for patients with these two specific scenarios for HCV elimination [17,18,19]. Thus, this study aimed to evaluate the real-world safety and efficacy of pan-genotypic DAA in patients with mixed-genotype or genotype-undetermined HCV from the five hospitals in the Changhua Christian Care System.

## 2. Materials and Methods

### 2.1. Materials

This retrospective study included DAA treatment-naïve patients undergoing treatment for HCV infection with either mixed or undetermined HCV genotypes, who received ≥1 dose of DAA between August 2018 and December 2020 at five hospitals of the Changhua Christian Health Care System. The study was approved by the Changhua Christian Hospital Institutional Review Board (CCH IRB No 210202), and the requirement for participants’ informed consent was waived because of the retrospective study design. Medical information, including demographics, baseline medical conditions, anti-HCV treatment regimen and duration, laboratory data, and information on adverse events were obtained from electronic patient records. All procedures were carried out in accordance with relevant guidelines and regulations of the National Health Insurance Administration of Taiwan.

### 2.2. Treatment, Efficacy, and Safety Evaluation

Our primary goal was to evaluate the treatment result of pan-genotype DAA for patients with HCV infection. We used ART HCV assays (RealTime HCV and HCV Genotype II, Abbott Molecular, Abbott Park, IL, USA) to quantify HCV viral RNA load and genotyping. The assay detects genotypes 1, 2, 3, 4, 5, and 6, and subtypes 1a and 1b with genotype-specific fluorescent-labeled oligonucleotide probes [20,22,23]. An “indeterminate” result means a detectable HCV viral load without the ability to produce a genotype result. The end-of-treatment viral response (ETVR) was defined as an HCV RNA level that was less than the lower limit of quantification (LLOQ) when completing the treatment course. An SVR was defined as an HCV RNA level that was less than the LLOQ at 12 weeks after the last medication. The treatment regimen of each DAA was prescribed according to the drug label. The SOF/VEL group underwent a twelve-week therapy and GLE/PIB group underwent either an eight-week or a 12-week therapy. Virologic treatment failure was defined as either (a) non-response: HCV was detected during and at end of the treatment; or (b) relapse: HCV was undetectable at the end of the treatment but detectable during the follow-up period. Two endpoints for SVR were evaluated. The intention-to-treat group (ITT) included patients receiving at least one dose of DAA and the per-protocol group (PP) was established by excluding patients due to non-virological failure.

### 2.3. Statistical Analyses

Baseline data were analyzed to compare two HCV genotype groups (Mixed type/Indeterminate type) using Student’s *t*-test or the Mann–Whitney U test for continuous data and using the chi-square test or Fisher’s exact test for categorical data. The distribution of continuous variables was checked using the One-sample Kolmogorov–Smirnov Test. Statistical analyses were performed using IBM SPSS version 22.0 (IBM Corp., Armonk, NY, USA) and MedCalc statistical software MedCalc Version 19.8 (MedCalc Software Ltd. Acacialaan 22, 8400 Ostend, Belgium). The results were considered statistically significant if the two-tailed *p*-value was <0.05 for all tests.

## 3. Results

### 3.1. General Characteristics of the Study Population

A total of 2446 HCV-infected patients underwent pan-genotype anti-HCV therapy, including GLE/PIB (n = 1527) and SOF/VEL (n = 919) during the study period (Figure 1 and Table 1). The genotype of study population includes single-genotype infection with type 1 (48.9%) followed by type 2 (36.6%), type 6 (4.5%), type 3 (2.9%), and type 4 (0.04%). Thirty-seven (1.5%) patients had mixed-genotype HCV infection and 110 (4.5%) had undetermined HCV genotypes. Majority of the patients were male (55.8%), and the mean age of the study participants was 63 years. Three percent of the patient had prior interferon failure or interruption. Cirrhosis was present in 13.7% of the study participants. There is no statistically significant difference in patient characteristics between the two patient populations (Table 1). As shown in Table 2, the three most common mixtures of genotypes were 1b + 2 (45.9%), 1 + 6 (18.9%), and 1 + 3 (8.1%).

### 3.2. Treatment Effectiveness

One patient discontinued the therapy due to spontaneous bacterial infection with mortality. Three fatalities from pneumonia, aortic dissection, and cholangitis were observed after the treatment. All the deaths were judged not to be associated with the treatment regimen. Two patients were lost to follow-up. Five patients from the GLE/PIB treated group who did not achieve ETVR finally achieved SVR at the end of 12-week follow-up (SVR+) after therapy (Figure 1). These five patients have a low HCV viral load at the end of treatment (ranges from 12 IU/mL to 36 IU/mL. The overall ITT SVR rate was 92.5%, and the PP SVR rate was 96.5% (Table 3). We found no statistically significant difference in the SVR rate between the patients with the mixed-genotype infection and those with the undetermined-genotype infection.

### 3.3. Laboratory Adverse Events of the Treatment

During the treatment period, patients received laboratory tests for liver function and blood cell count during each outpatient visit. A significant increase (≥5× elevation) in the levels of bilirubin, GOT, and GPT was observed in 0.7% of the study participants (Table 4). Grade 3 anemia was observed in 2.5% of the study participants and grade 3 thrombocytopenia was observed among 2.5% of the patients. There was no side-effect-related premature treatment termination in any of the study participants.

## 4. Discussion

In this multiple-center cohort study of mixed-genotype or genotype-undetermined HCV-infected patients receiving pan-genotypic DAA therapy, we found an overall SVR rate of 96.5% via PP analysis in a real-world setting. Despite the well-known treatment efficacy of these pan-genotypic DAAs in their landmark clinical trial setting [24,25,26,27], data on the effect of DAA treatment for these less frequently encountered HCV genotype populations are scarce [17,28,29]. In this study, we attempted to fill this knowledge gap on the DAA treatment of infection and demonstrated that the pan-genotypic DAA with either SOF/VEL or GLE/PIB are effective and safe for these patients in the real-world setting. Our results further support the fact that the simplified treatment algorithm by the AASLD-IDSA guideline [16] could efficiently overcome the HCV elimination barrier in our daily medical practice.

The HCV could be classified into six main genotypes based on the sequences of the viral genome with differences of 30–35% in their nucleotide sequences [30,31]. The geographic distribution of HCV genotypes varies widely according to historical events or human migration trends. While genotype 1 was found worldwide, genotype 4 was found mainly in North Africa and the Middle East while genotype 6 was found in Southeast Asia [31]. Genotype 3 was commonly found in drug abusers. Determination of HCV genotype requires the use of PCR to hybridize to genotype-specific probe that allows understanding route of HCV infection of the patient and helps clinicians to determine the duration of interferon therapy [2,6] or choosing genotype-specific DAA therapy before the introduction of current pan-genotypic DAA therapy [14,18,19,32]. However, less than ten percent of the patients were infected with the mixed variant of the HCV, especially those infected with HIV or intravenous drugs users [17,21,33,34,35]. In this study, high rates of genotype mixtures of 1 + 6 (18.9%) were interesting because genotypes 1b and 2 were the most prevalent in Taiwan. Such a finding could be explained by the high-risk behaviors of the patients at some time of their lives in Southeast Asia [36]. The HCV genotype could be found to be undetermined by the commercial assay while the viral load was below the detection limit that requires further genetic sequencing [19,21]. Although the universal use of gene sequencing allows for more delicate genotyping, such an approach has not yet been standardized. The routine use of this approach is not feasible in daily clinical practice.

To achieve the goal set by the World Health Organization for HCV elimination, a diagnostic rate of 90% and a treatment coverage rate of 80% are required to achieve a 65% reduction in the rate of HCV-related deaths by 2030 [4,10,34,35,37]. A major challenge is determining the HCV genotype before initiating DAA regimens. This is of particular importance in resource-limited countries, high-prevalence populations such as PWID, or people in outreach onsite treatment programs such as prisons [38]. Since the introduction of pan-genotypic DAA regimens, several society guidelines have removed the requirement for HCV genotype documentation before prescribing such DAAs [39] due to their high efficacy. Such a simplified treatment pathway is particularly beneficial for people in the PWID population, homeless patients, migrants, and incarcerated people in the era of the COVID-19 pandemic. Recently, Chiu et al. [36] reported an SVR rate that ranged from 96.6% to 100% among 116 patients and Ding et al. [17] reported a 100% SVR rate among 108 patients with mixed infections who were treated with pan-genotypic DAAs. Our satisfactory result of the treatment includes not only mixed-genotype variants but also genotype-undetermined variants, which further disparages the requirement for stringent genotype determination to facilitate HCV elimination.

The present study has several limitations. First, it was a retrospective study that was subject to reporting bias concerning the side effects of the treatment. Second, our study included only naïve patients undergoing DAA treatment. Therefore, we cannot extrapolate the efficacy data to patients who have undergone prior failed DAA therapy. Third, because genotype 5 is rare in Taiwan, this study did not include patients with mixed infections that involved genotype 5. Finally, the genotype reports from our laboratory were not further confirmed by genetic sequencing. As mentioned, no consensus on the standardized method for the detection of mixed-genotype or genotype-undetermined HCV has been obtained in routine clinical practice.

## 5. Conclusions

In this real-world study, we found that current pan-genotypic DAAs were effective and well-tolerated for mixed-genotype or genotype-undetermined HCV infection.

## Figures and Tables

**Figure 1 jcm-11-01853-f001:**
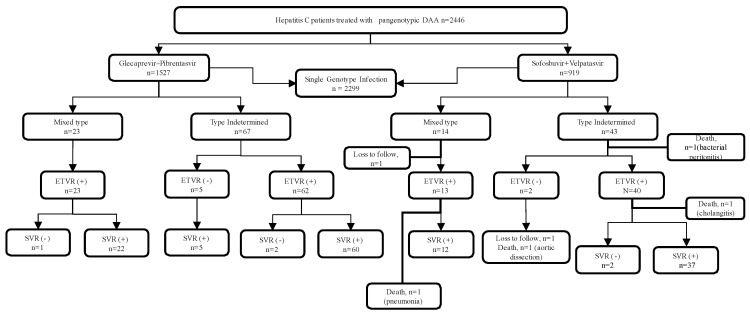
Flowchart of the study population.

**Table 1 jcm-11-01853-t001:** Patient characteristics.

Parameter	All Patients	Mixed Type	Indeterminate Type	*p*-Values
Gender (Male), n/N (%)	82/147 (55.8%)	24/37 (64.9%)	58/110 (52.7%)	0.198
Age, yr, mean ± SD	63 ± 13	63 ± 12	63 ± 14	0.912
Body Height, cm, mean ± SD	160.5 ± 8.8	161.4 ± 9.6	160.2 ± 8.5	0.494
Body weight, kg, mean ± SD	64.8 ± 12.6	65.4 ± 10.6	64.6 ± 13.2	0.779
Body Mass Index, kg/m^2^, mean ± SD	24.9 ± 3.9	24.7 ± 3.6	25 ± 4	0.737
Comorbidity, n/N (%)				
Hepatoma	3/147 (2.0%)	2/37 (5.4%)	1/110 (0.9%)	0.156
Cirrhosis	20/147(13.7%)	3/37 (8.1%)	17/110 (15.5%)	0.252
Renal Failure	9/147 (6.1%)	2/37 (5.4%)	7/110 (6.4%)	1.000
PWID	5/147 (3.4%)	1/37 (2.7%)	4/110 (3.6%)	1.000
DM	28/147 (19.0%)	7/37 (18.9%)	21/110 (19.1%)	0.982
HTN	46/147 (31.3%)	10/37 (27.0%)	36/110 (32.7%)	0.518
HBV	9/147 (6.1%)	2/37 (5.4%)	7/110 (6.4%)	1.000
Previous Interferon Therapy, n/N (%)				0.129
Prior interferon failure	3/147 (2.0%)	2/37 (5.4%)	1/110 (0.9%)	
No interferon therapy	142/147 (96.6%)	34/37 (91.9%)	108/110 (98.2%)	
Prior interferon interruption	2/147 (1.4%)	1/37 (2.7%)	1/110 (0.9%)	
Regimen of DAA, n/N (%)				0.892
Glecaprevir–Pibrentasvir	90/147 (61.2%)	23/37 (62.2%)	67/110 (60.9%)	
Sofosbuvir + Velpatasvir	57/147 (38.8%)	14/37 (37.8%)	43/110 (39.1%)	
Laboratory Data				
HCV viral load, IU/mL, median (IQR)	1,257,396 (71,390–4,118,819)	1,236,277 (157,609–2,996,333)	1,2768,32 (39,167–4,504,185)	0.810
AST, U/L, median (IQR)	37 (27–53)	36 (29–49)	38 (27–56)	0.806
ALT, U/L, median (IQR)	43 (27–69)	44 (29–66)	43 (27–69)	0.603
Platelet count, ×10^3^/μL, mean ± SD	188 ± 67	193 ± 69	186 ± 67	0.595
Hb, g/dL, median (IQR)	13.7 (12–14.7)	14.1 (13–15.1)	13.6 (12–14.4)	0.063
I.N.R., median (IQR)	0.99 (0.95–1.04)	1.01 (0.95–1.06)	0.99 (0.95–1.04)	0.955
Bilirubin, mg/dL, median (IQR)	0.64 (0.5–0.87)	0.7 (0.5–0.9)	0.62 (0.5–0.8)	0.280
Albumin, g/dL, median (IQR)	3.9 (3.7–4.2)	3.9 (3.7–4.1)	4 (3.7–4.2)	0.616
Creatinine, mg/dL, median (IQR)	0.9 (0.69–1.07)	0.92 (0.69–1.17)	0.9 (0.7–1.07)	0.996
FIB4, median (IQR)	2.08 (1.34–3.34)	2.06 (1.39–3)	2.15 (1.34–3.45)	0.653
APRI, median (IQR)	0.513 (0.321–0.921)	0.514 (0.348–0.87)	0.512 (0.317–0.94)	0.930

Abbreviations: ALT—alanine aminotransferase; APRI—AST to Platelet Ratio Index; DM—diabetes mellitus; DAA—direct antiviral agent; FIB-4—Fibrosis-4; HBV—hepatitis B virus; HTN—hypertension; I.N.R.—international normalized ratio; IQR: interquartile range; PWID—person who inject drugs.

**Table 2 jcm-11-01853-t002:** Frequency of mixed hepatitis C virus genotypes.

HCV Genotype, n (%)	Total of Mixed Type (N = 37)	GLE/PIB (N = 23)	SOF/VEL (N = 14)
1 + 2 + 6	2/37 (5.4%)	2/23 (8.7%)	0/14 (0.0%)
1 + 3	3/37 (8.1%)	2/23 (8.7%)	1/14 (7.1%)
1 + 4	2/37 (5.4%)	0/23 (0.0%)	2/14 (14.3%)
1 + 6	7/37 (18.9%)	5/23 (21.7%)	2/14 (14.3%)
1a + 2	1/37 (2.7%)	0/23 (0.0%)	1/14 (7.1%)
1a + 4	1/37 (2.7%)	0/23 (0.0%)	1/14 (7.1%)
1b + 2	17/37 (45.9%)	11/23 (47.8%)	6/14 (42.9%)
1b + 3	2/37 (5.4%)	2/23 (8.7%)	0/14 (0.0%)
3 + 4	2/37 (5.4%)	1/23 (4.3%)	1/14 (7.1%)

**Table 3 jcm-11-01853-t003:** Treatment response of the study population.

HCV RNA < LLOQ	All Patients (n = 147)		Mixed Type (n = 37)		Undetermined (n = 110)		*p*-Values
	n/N (%)	95% CI	n/N (%)	95% CI	n/N (%)	95% CI	
During treatment							
ETVR (ITT)	138/147 (93.9%)	88.7–97.2	36/37 (97.3%)	85.8–99.9	102/110 (92.7%)	86.2–96.8	0.450
ETVR (PP)	138/145 (95.2%)	90.3–98.0	36/36 (100.0%)	90.3–100	102/109 (93.6%)	87.2–97.4	0.193
After treatment							
SVR_12_ (ITT)	136/147 (92.5%)	87.0–96.2	34/37 (91.9%)	78.1–98.3	102/110 (92.7%)	86.2–96.8	1.000
SVR_12_ (PP)	136/141 (96.5%)	91.9–98.8	34/35 (97.1%)	85.1–99.9	102/106 (96.2%)	90.6–99.0	1.000

LLOQ—lower limit of quantification is 12 IU/mL; ETVR—end of treatment virological response; SVR_12_—sustained virologic response rate at off-treatment week 12; ITT—intention-to-treat; PP—per-protocol population.

**Table 4 jcm-11-01853-t004:** Laboratory side effects during therapy.

	All Patients	Mixed Type	Indeterminate Type	*p*-Values
GPT, n/N (%)				0.441
<3× elevation	145/147 (98.6%)	36/37 (97.3%)	109/110 (99.1%)	
3–5× elevation	2/147 (1.4%)	1/37 (2.7%)	1/110 (0.9%)	
≥5× elevation	0/147 (0.0%)	0/37 (0.0%)	0/110 (0.0%)	
GOT, n/N (%)				0.252
<3× elevation	146/147 (99.3%)	36/37 (97.3%)	110/110 (100.0%)	
3–5× elevation	1/147 (0.7%)	1/37 (2.7%)	0/110 (0.0%)	
≥5× elevation	0/147 (0.0%)	0/37 (0.0%)	0/110 (0.0%)	
Bilirubin, n/N (%)				0.799
<1.5× elevation	134/147 (91.2%)	35/37 (94.6%)	99/110 (90.0%)	
1.5–3× elevation	12/147 (8.2%)	2/37 (5.4%)	10/110 (9.1%)	
≥3× elevation	1/147 (0.7%)	0/37 (0.0%)	1/110 (0.9%)	
Anemia, n/N (%)				0.756
G0 *	84/120 (70.0%)	22/30 (73.3%)	62/90 (68.9%)	
G1	27/120 (22.5%)	6/30 (20.0%)	21/90 (23.3%)	
G2	6/120 (5.0%)	2/30 (6.7%)	4/90 (4.4%)	
G3	3/120 (2.5%)	0/30 (0.0%)	3/90 (3.3%)	
Thrombocytopenia, n/N (%)				0.660
G0	83/119 (69.7%)	23/30 (76.7%)	60/89 (67.4%)	
G1	29/119 (24.4%)	5/30 (16.7%)	24/89 (27.0%)	
G2	4/119 (3.4%)	1/30 (3.3%)	3/89 (3.4%)	
G3	3/119 (2.5%)	1/30 (3.3%)	2/89 (2.2%)	

* Graded according to Common Terminology Criteria for Adverse Events version 5.0.

## Data Availability

The analyzed data are available on reasonable request.

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
