# Peer review of "Pan-Genotypic Direct-Acting Antiviral Agents for Undetermined or Mixed-Genotype Hepatitis C Infection: A Real-World Multi-Center Effectiveness Analysis"

_jcm, 2022, doi:10.3390/jcm11071853_

Round 1

Reviewer 1 Report

Manuscript titled “Pan-genotypic direct-acting antiviral agents for undetermined or mixed-genotype hepatitis C infection: A real-world multi-center effectiveness analysis” by Yen et al is well structured, the data is nicely interpreted and written clearly. Based on the analysis and interpretation, Authors concluded that irrespective of the HCV genotypes, Pan-genotypic direct-acting antiviral agents are effective and well-tolerated for treating HCV patients. My comments are provided below.

Major comment

5 Patients whose genotype was undetermined from GLE/PIB treated group, showed no response to therapy (ETVR-) (i.e., HCV was detected) at the end of the treatment (Figure 1). However, their HCV RNA viral load is undetectable at the end of 12-week follow-up (SVR+) after therapy (Figure 1). Is this observation being correct? If so, please discuss in detail about this, also provide the viral load for those patients at the end of treatment? Or, if the observation is incorrect and this is just a typo error, please re-do the mathematics in table 3 and correct the numbers accordingly in the manuscript wherever required.

Minor comments

In abstract, line 32: Mention the name of the treatment regimens in closed bracket.

In abstract, Authors mentioned “We lost one patient to follow during the treatment and four died”. However, in figure 1 and line 155, Authors mentioned “Two patients were lost to follow-up. Based on the Figure 1, Authors lost one patient for follow-up during treatment and one more patient after treatment. Please correct the abstract accordingly.

Line 97: Correct the sentence from “HCV RNA concentrations” to “HCV viral RNA load/level/copy number”

Follow the term, HCV throughout the manuscript instead of using the term, hepatitis C.

Line 160: Correct the word, “qualification” into “Quantification”.

Please correct the sentence, “Most patients were male” to “Majority of the patients were male”.

Mention, Gender (Male) in Table 1 instead of mentioning Gender alone.

If you could explain, please discuss what makes those 5 patients (SVR-) showed virologic non-response to therapy during/after 12 weeks of follow-up?

Author Response

Dear Reviewer,

Thank you for reviewing our manuscript and providing your editorial comments as well as reviewers comments for improving our manuscript. Based on these comments, we have made several revisions to our manuscript, which we are hereby resubmitting for your consideration. Our point-by-point responses to the comments are detailed below.

Response to Reviewers’ comments

#1. Major comment

5 Patients whose genotype was undetermined from GLE/PIB treated group, showed no response to therapy (ETVR-) (i.e., HCV was detected) at the end of the treatment (Figure 1). However, their HCV RNA viral load is undetectable at the end of 12-week follow-up (SVR+) after therapy (Figure 1). Is this observation being correct? If so, please discuss in detail about this, also provide the viral load for those patients at the end of treatment? Or, if the observation is incorrect and this is just a typo error, please re-do the mathematics in table 3 and correct the numbers accordingly in the manuscript wherever required.

Response: Thank for your comment. The 5 patients have a low detectable HCV RNA in the ERVR point ( HCV RNA:35,30,16,12, and 12 IU respectively ) and became undetectable at SVR point.

#2. In abstract, line 32: Mention the name of the treatment regimens in closed bracket.

Response: Thank for your comment. We made revision according to your suggestion.

#2. In abstract, Authors mentioned “We lost one patient to follow during the treatment and four died”. However, in figure 1 and line 155, Authors mentioned “Two patients were lost to follow-up. Based on the Figure 1, Authors lost one patient for follow-up during treatment and one more patient after treatment. Please correct the abstract accordingly.

Response: Thank for your comment. We made revision according to your suggestion.

#4. Line 97: Correct the sentence from “HCV RNA concentrations” to “HCV viral RNA load/level/copy number”

Response: Thank for your comment. We made revision according to your suggestion.

#5. Follow the term, HCV throughout the manuscript instead of using the term, hepatitis C.

Response: Thank for your comment. We made revision according to your suggestion.

#6. Line 160: Correct the word, “qualification” into “Quantification”.

Response: Thank for your comment. We made revision according to your suggestion.

#7. Please correct the sentence, “Most patients were male” to “Majority of the patients were male”.

Response: Thank for your comment. We made revision according to your suggestion.

#8. Mention, Gender (Male) in Table 1 instead of mentioning Gender alone.

Response: Thank for your comment. We made revision according to your suggestion

#9. If you could explain, please discuss what makes those 5 patients (SVR-) showed virologic non-response to therapy during/after 12 weeks of follow-up?

Response: Thank for your comment. As mentioned previously, these 5 patients have a very low HCV RNA at the ETVR point and therefore have SVR during the follow-up period.

Thank you for the opportunity to resubmit this manuscript for consideration for publication in the Medicine If you have any questions or comments regarding this manuscript, please do not hesitate to contact us using the details provided below.

Sincerely,

Hsu-Heng Yen, M.D

Pei-Yuan Su, M.D

Division of Gastroenterology, Department of Internal Medicine, Changhua Christian Hospital, Changhua, Taiwan

Fax: +886-4-7228289

Tel: +886-4-7238595ext5501

E-mail: [email protected], [email protected]

Reviewer 2 Report

The authors performed a retrospective study, which evaluates the efficacy and safety of the pan-genotypic direct-acting antiviral (DAAs) treatment in HCV-infected adults who had mixed or undetermined HCV genotypes. In this multiple-center cohort study, these patients were from five hospitals in the Changhua Christian Care System in Taiwan. Although there were several limitations, as mentioned in the discussion, the clinical results showed that pan-genotypic DAAs (SOF/VEL or GLE/PIB) were effective and safe for mixed-genotype or genotype-undetermined HCV infection. The results are not surprising. Still, this study enriches the real-world data regarding the efficacy and safety of DAA treatment against HCV infection. The manuscript is well-written and easy to follow. It should be minor changes. For instance, the text in line 137 seems wrong.

Author Response

Dear Reviewer,

Thank you for reviewing our manuscript and providing your editorial comments as well as reviewers comments for improving our manuscript. Based on these comments, we have made several revisions to our manuscript, which we are hereby resubmitting for your consideration. Our point-by-point responses to the comments are detailed below.

Response to Reviewers’ comments

The authors performed a retrospective study, which evaluates the efficacy and safety of the pan-genotypic direct-acting antiviral (DAAs) treatment in HCV-infected adults who had mixed or undetermined HCV genotypes. In this multiple-center cohort study, these patients were from five hospitals in the Changhua Christian Care System in Taiwan. Although there were several limitations, as mentioned in the discussion, the clinical results showed that pan-genotypic DAAs (SOF/VEL or GLE/PIB) were effective and safe for mixed-genotype or genotype-undetermined HCV infection. The results are not surprising. Still, this study enriches the real-world data regarding the efficacy and safety of DAA treatment against HCV infection. The manuscript is well-written and easy to follow. It should be minor changes. For instance, the text in line 137 seems wrong.

Response: Thank for your comment. We made revision to the Figure 1 tittle in the revised manuscript.

Thank you for the opportunity to resubmit this manuscript for consideration for publication in the Medicine If you have any questions or comments regarding this manuscript, please do not hesitate to contact us using the details provided below.

Sincerely,

Hsu-Heng Yen, M.D

Pei-Yuan Su, M.D

Division of Gastroenterology, Department of Internal Medicine, Changhua Christian Hospital, Changhua, Taiwan

Fax: +886-4-7228289

Tel: +886-4-7238595ext5501

E-mail: [email protected], [email protected]

Round 2

Reviewer 1 Report

Thank you for your point-by-point response.